# Molecular Responses to Thermal and Osmotic Stress in Arctic Intertidal Mussels (*Mytilus edulis*): The Limits of Resilience

**DOI:** 10.3390/genes13010155

**Published:** 2022-01-15

**Authors:** Nicholas J. Barrett, Jakob Thyrring, Elizabeth M. Harper, Mikael K. Sejr, Jesper G. Sørensen, Lloyd S. Peck, Melody S. Clark

**Affiliations:** 1British Antarctic Survey, Natural Environment Research Council, Cambridge CB3 0ET, UK; emh21@cam.ac.uk (E.M.H.); lspe@bas.ac.uk (L.S.P.); mscl@bas.ac.uk (M.S.C.); 2Department of Earth Sciences, University of Cambridge, Cambridge CB2 3EQ, UK; 3Department of Ecoscience—Marine Ecology & Arctic Research Centre, Aarhus University, 8000 Aarhus C, Denmark; thyrring@ecos.au.dk (J.T.); mse@ecos.au.dk (M.K.S.); 4Department of Zoology, University of British Columbia, Vancouver, BC V6T 1Z4, Canada; 5Department of Biology—Genetics, Ecology and Evolution, Aarhus University, 8000 Aarhus C, Denmark; jesper.soerensen@bio.au.dk

**Keywords:** blue mussel, cellular stress response, salinity, thermal tolerance, transcriptome, acclimation, freshening, climate change, aquaporins

## Abstract

Increases in Arctic temperatures have accelerated melting of the Greenland icesheet, exposing intertidal organisms, such as the blue mussel *Mytilus edulis,* to high air temperatures and low salinities in summer. However, the interaction of these combined stressors is poorly described at the transcriptional level. Comparing expression profiles of *M. edulis* from experimentally warmed (30 °C and 33 °C) animals kept at control (23‰) and low salinities (15‰) revealed a significant lack of enrichment for Gene Ontology terms (GO), indicating that similar processes were active under all conditions. However, there was a progressive increase in the abundance of upregulated genes as each stressor was applied, with synergistic increases at 33 °C and 15‰, suggesting combined stressors push the animal towards their tolerance thresholds. Further analyses comparing the effects of salinity alone (23‰, 15‰ and 5‰) showed high expression of stress and osmoregulatory marker genes at the lowest salinity, implying that the cell is carrying out intracellular osmoregulation to maintain the cytosol as hyperosmotic. Identification of aquaporins and vacuolar-type ATPase transcripts suggested the cell may use fluid-filled cavities to excrete excess intracellular water, as previously identified in embryonic freshwater mussels. These results indicate that *M. edulis* has considerable resilience to heat stress and highly efficient mechanisms to acclimatise to lowered salinity in a changing world.

## 1. Introduction

The thermal response of organisms is one of the best quantified in biology (e.g., Clarke and Johnston, 1999; Gillooly et al., 2001 [1,2]) and is essential for predicting the effects of climate change [3,4]. However, knowledge of how additional stressors modify organismal thermal tolerance is still surprisingly limited. In recent years, the Arctic has experienced a rate of warming that is three times higher than the global average [5]. In Greenland, warmer air temperatures have increased melting from glaciers and land ice [6,7], contributing to increased coastal freshwater input and observed declines in sub-surface salinity levels [8]. This has direct consequences for the inhabitants of nearshore ecosystems, especially intertidal organisms, which face the dual challenge of increasingly warmer air temperatures and fresher coastal waters under current and future climate scenarios [9,10].

The blue mussel, *Mytilus edulis*, is a boreal bivalve mollusc that has a distribution stretching from warm temperate to polar latitudes. It is a common inhabitant of the intertidal benthic communities of west Greenland fjords [11] and is exposed to a wide range of air temperatures and salinities. For example, *M. edulis* exhibits freeze-tolerant abilities [12,13], with a lower lethal limit (LLT_50_) of −13 °C measured in populations from west Greenland [14]. At the opposite end of the scale, high intertidal air temperatures of ~36 °C are becoming more common in the Greenland intertidal zone [15]; however, recent studies have demonstrated a remarkable resilience to high air temperatures in *M. edulis* [14,16,17]. Upper lethal temperatures for *M. edulis* have been observed to be between 25 °C and 38 °C [14,18,19,20]. This high degree of thermal tolerance in *Mytilus* species is largely dependent on environmental conditions, with acclimation and the predictability of thermal stress being considered key modifiers of thermal resilience [17,21].

With regards to salinity tolerance, *M. edulis* are euryhaline osmoconformers [22], with extracellular fluids being isosmotic with their surrounding medium [23]. Their haline range extends from full-strength seawater (35‰) to very low salinity (~5‰) brackish waters [19,24], although they can tolerate salinities close to zero for limited periods of time [25,26,27]. Short-term low salinity tolerance in *M. edulis* is initially possible through behavioural mechanisms, which can include closing their shell valves and the exhalant siphon on instant exposure to lowering salinity water [26]. This effectively traps water within the mantle cavity, which can be maintained at a higher osmolality than the external seawater [28]. However, build-up of metabolic waste products, in addition to respiratory and feeding requirements, eventually encourages valve gaping. This brings the mantle tissue in direct contact with external water, initiating osmotic rebalancing of the haemolymph. In turn, this triggers cell volume regulatory changes, which in molluscs constitutes an efflux of intracellular organic osmolytes across the cell membrane [29,30,31], allowing cells to adjust their osmotic concentration to match that of the extracellular fluid. These cellular adjustments mark the beginning of the acclimatisation process, considered to take one to four weeks in laboratory conditions [32]. Both short and long-term salinity-tolerance strategies come at a considerable metabolic cost, and consequently mussels living in lower salinity environments have higher respiration rates [33], are smaller [34,35], have thinner shells [36] and exhibit depressed shell growth rates [37,38].

Temperature and salinity are considered “master factors” with regard to development, growth and survival of marine animals [39]. As such, the combined effects of temperature and salinity may interact in a complex, non-linear manner, producing responses that would not have been predicted from observing a single variable in isolation [40]. Low salinity has already been demonstrated to significantly depress the temperature at which mortality occur in *M. edulis* from west Greenland [16], suggesting a synergistic interaction between temperature and salinity. At the molecular level, abrupt environmental change often involves the production of stress proteins, such as heat shock proteins (HSPs) [41]. For *Mytilus* species, upregulation of HSPs has been observed in response to heat stress [16,42,43] and osmotic stress [16,42,44]. Previous studies comparing differential gene expression in *Mytilus* species under both high temperature and low salinity conditions observed only a small number of overlapping genes, which showed divergent expression patterns [43,44]. However, Nielsen et al. (2021) [16] observed upregulation of *HSP70* under both low salinity and heat stress.

This study aims to build on findings from a previous ecophysiological study [16] to further explore the molecular responses of *M. edulis* to thermal and osmotic stress by using an RNA-Seq approach. This technique offers detailed molecular insight into the direct responses to environmental stressors by quantifying the presence of transcribed RNA molecules at the point of sampling. A subset of samples generated by Nielsen et al. (2021) [16] where mean mortality was <25% have been used in the RNA-Seq analyses in this study. These samples were partitioned into two separate analyses, with 23‰ at 5 °C used as a control in each experiment:

(1) Low salinity exposure: using samples held at 23‰, 15‰ and 5‰, all at 5 °C. A focus of this analysis was to investigate the physiological mechanisms that mussels employ at very low salinities by analysing the regulation of key marker genes previously identified as important in bivalve osmoregulation including ion channel, aquaporin and osmolyte transporter genes [44,45].

(2) Heat stress and low salinity exposure: the interaction between temperature and salinity using samples held at 23‰ and 15‰, subjected to air exposures of 30 °C and 33 °C. This analysis focused on the interaction between the two stressors and interrogated the extent of the molecular overlap and energy resource allocation.

## 2. Materials and Methods

### 2.1. Specimen Collection

Blue mussels (*M. edulis*) were collected from the lower intertidal zone of Kobbefjord south of Nuuk, Greenland (64°10′50.8″ N, 51°32′28.1″ W) in August 2019. Mussels were kept in seawater obtained from the collection site and held at 5 °C for 3 days before transfer to a holding facility at Aarhus University, Denmark. They were then divided into four aquaria with daily seawater changes, in which water temperature was maintained at 5 °C and salinity at ~23‰, corresponding to the salinity of the collection site. The mussels were exposed to 5 °C air twice a day for 1.5 h at 12 h intervals (simulating a semi-diurnal tide) for one week, prior to starting experimentation to acclimate the animals to the new experimental conditions.

### 2.2. Salinity and Temperature Experiments

Mussels were distributed among 48, 0.5 L buckets (*n* = 10 per bucket; total mussels *n* = 480). The buckets had holes drilled to allow water to pass through exposing the mussels to air, when simulating a low tide. The buckets were distributed between 12 aquaria, four for each of the three salinity conditions (5‰, 15‰ and 23‰). Within each salinity treatment, mussels were exposed to experimental air temperatures of 5 °C, 30 °C and 33 °C every 12th hour for 1.5 h over a 6-day period. As *M. edulis* has an upper temperature limit of 36 °C, temperature values close to the upper limit were chosen. Each treatment combination had four replicate buckets, totalling 40 mussels per treatment combination. For a detailed methodological description of the mussel sampling procedure and physiological experimental design, see Nielsen et al. (2021) [16].

At the end of the experiment, mussel length was measured, and gill tissue dissected. Gill samples were flash-frozen and stored at −80 °C until use.

### 2.3. RNA Extraction and Sequencing

Prior to selecting samples for RNA extraction, tank effects were tested on mortality data using a Fischer’s exact test of independence on any tank where mortality was observed (four tanks). There were no tank effects (*p* > 0.6), and hence animals were sampled from each replicate tank for each treatment. A number of treatments from Nielsen et al. (2021) [16] have been excluded. The 36 °C air temperature treatment resulted in 100% mortality, while mortality in mussels exposed to 5‰ salinity and subjected to heat stress (both 30 °C and 33 °C) was >50%, representing a median lethal mortality (LD50). Seven treatments were selected for transcriptomic analysis, based on the previous results of Nielsen et al. (2021) [16], and comprised the following conditions, hereafter referred to by the name in brackets: 23‰ at 5 °C (control), 23‰ at 30 °C (30 °C), 23‰ at 33 °C (33 °C), 15‰ at 5 °C (low salinity), 15‰ at 30 °C (low salinity +30 °C), 15‰ at 33 °C (low salinity +33 °C) and 5‰ at 5 °C (very low salinity) (*n* = 5 per treatment; *n* = 35 samples in total). Mussel shell length was used as the selection criterion for choosing samples to extract RNA. Mussels with shell lengths as near as possible to the mean of all samples (34.7 ± 0.13 mm) were chosen to ensure the animals sampled were of a similar age. This represented one from each of the four tank replicates per treatment plus a fifth based on the nearest suitable size. For each sample, ~30 mg of gill tissue was extracted using the SV total RNA isolation system (Promega, Madison, WI, USA) (which includes a DNase treatment), according to manufacturers’ instructions. All samples were checked for RNA purity and concentration on a NanoDrop™ (ThermoFisher, Waltham, MA, USA) spectrophotometer and an Agilent TapeStation™ (Agilent, Santa Clara, CA, USA). Thirty-five individual samples with an RNA integrity number (RIN) ≥8 were sent for library preparation and NovaSec 6000 PE150 (Illumina, San Diego, CA, USA) sequencing at Novogene (Cambridge, UK).

### 2.4. Bioinformatics

The raw sequencing reads from the 35 samples were subjected to quality control, transcript assembly and downstream analysis. This was performed by NovoGene via the following bioinformatics pipeline. Raw-read data were quality-controlled for error rate distribution using Phred score and GC content distribution. Data filtering removed Illumina adaptor sequences, low-quality reads with uncertain nucleotides (*N* > 10%) and reads with low quality bases that constituted >50% of the read. Transcriptome assembly using the cleaned reads was carried out *de novo* using Trinity 2.6.6 (with parameters minKmerCov = 3 and min_glue = 4), and utilising modules Inchworm, Chrysalis and Butterfly [46]. Hierarchical clustering was performed using Corset 4.6 [47], with parameters set to default (except; -f ture, -m 10), whose groups contigs were based on shared reads, while contigs differentially expressed between treatments were also separated, ultimately removing read redundancy. The longest transcripts from each cluster were selected as unigenes, and these were subjected to gene annotation. Seven gene databases (NR, NT, KEGG, SwissProt, Pfam, GO and KOG; see Table 1) were utilised to achieve functional gene annotation for the unigenes. For each database, the following software and parameters were implemented: NCBI blast 2.9.0 with an e-value threshold of 1 × 10^−5^ was performed against the NT database [48]. Diamond 0.8.22 [49] with an e-value threshold of 1 × 10^−5^ was used to blast unigenes against the NR, KOG and Swiss-Prot databases. HMMER 3.1, utilising the hmmscan package with an e-value threshold of 0.01, was used to search the Pfam database [50]. GO annotation was achieved utilising the protein annotation results from NR and Pfam within Blast2GO b2g4pipe_v2.5 [51] software with an e value threshold of 1 × 10^−6^. KEGG annotation [52,53] was carried out using KASS (KEGG Automatic Annotation Server) [54] with an e-value threshold of 1 × 10^−5^. GO enrichment was performed using GOseq 1.32.0 [55] and topGO 2.32.0 [56] with a corrected *p* value of <0.05. KEGG enrichment was performed using KOBAS 3.0 [52] with a corrected *p* value of <0.05.

For the gene-expression analysis, the de novo transcriptome was used as a reference sequence to map reads and quantify expression level using RSEM 1.2.28 (with parameters –estimate-rspd -mismatch-rate 0.3) [57] and Bowtie2 2.4.4 [58], outputting a read count of each transcript for each sample and then converting to a FPKM value (Fragments Per Kilobase per Million fragments mapped) [59]. The threshold for expression was set at FPKM >0.3. The differential expression analysis was carried out in DESeq2 1.26 [60] (which normalised read counts and estimated an FDR value using BH [61] with an adjusted *p* value of <0.05), and in edgeR 3.28 [62] with normalisation via TMM and an FDR value estimated using BH [61] with a log_2_fold change >1 and an adjusted *p* value of <0.005. The heatmap was generated in R 4.0.5 [63] within R Studio 1.1.456 [64] using the package pheatmap 1.0.12 [65], utilising FPKM values, which were log-transformed to account for zeros (e.g., log(count + 1)) and z-scored. Transcriptome data validation was undertaken by mapping the transcriptome onto a sample of primers (*HSP70* and *HSP90*) used in Nielsen et al. (2021) [16] in Geneious Prime (2022.0.1) [66].

Specific genes associated with the classical stress response identified in previous Greenland *Mytilus* studies [16,17], and those identified as being associated with osmoregulation in previous studies (e.g., Lockwood and Somero, 2011 [44]; Meng et al., 2013 [45]), were investigated in more detail via datamining of the differentially expressed gene lists. To determine the identity of putative aquaporin transcripts differentially expressed in the low salinity treatments (5‰ vs. control; 15‰ vs. control; 5‰ vs. 15‰), a phylogenetic tree was produced. A maximum likelihood phylogeny was generated in IQ-TREE 1.6.12 [67], using known aquaporin amino acid sequences from human, mouse, cow, fish and mollusc, in combination with the three translated aquaporin sequences from this study. For each representative species, amino acid sequences of *AQP0-12* were downloaded from GenBank, with the exception of *AQP2*, *AQP5* and *AQP6* in the zebrafish *Danio rerio* as tetrapod orthologues were absent; while in the scallop, *Mizuhopecten yessoensis*, aquaporin genes were predicted from genome analysis, so only a single sequence representative from each for the four aquaporin classes has been included (for accession numbers see; Appendix A). A 338-length amino acid sequence alignment was created using MAFFT 7.489 [68] using the L-INS-i method and subsequently hand-edited in Geneious Prime 2021.2.2 [66] to remove poorly aligned and divergent regions. Automatic model testing and selection was implemented within IQ-TREE using ModelFinder [69] with the LG + I + G4 model of sequence evolution selected based on the BIC scores. Bootstrap replicates (×1000) were performed within IQ-TREE using the ultrafast bootstrap [70].

## 3. Results

### 3.1. Transcriptome Statistics

RNA extracted from gill tissue samples from 35 individual *Mytilus edulis* was sequenced, generating over 913 million raw reads. After removing adaptor sequences and poor-quality reads, the transcriptome generated comprised 449,638 unigenes (Table 1). Trinity transcript and unigene figures were very similar (99.96%); therefore, we have only reported the unigene data. Library coverage across the 35 samples ranged from 18,925,398 to 32,269,107 clean reads with a Phred Q20 score ranging from 96.83% to 98.31% (Appendix A). Seven annotation databases were used to provide gene annotation, with the highest rates achieved using the NR (NCBI non-redundant protein sequences) database with a 35.08% annotation rate (Table 1). Due to the lack of availability of a full *M. edulis* genome for reference, the majority of annotations (>80%) comprised sequence similarities to four bivalve species: *Mytilus galloprovincialis* (27%), *Mizuhopecten yessoensis* (22.9%), *Magallana gigas* (previously reported as *Crassostrea gigas*) (16%), and *Crassostrea virginica* (14.8%), and one gastropod mollusc species (*Lottia gigantea* (1.7%)).

### 3.2. Differential Expression and GO Enrichment

The results of RNA-Seq analysis revealed high numbers of differentially expressed gene transcripts under all treatment comparisons ranging from 8515 to 17,998 (Table 2). Due to these large numbers, analyses concentrated on the exploitation of GO enrichment data and more detailed analyses of candidate genes putatively involved in osmoregulation and the stress response.

### 3.3. Low Salinity Exposure

These analyses comprised data generated from mussels exposed to different salinities (23‰, 15‰ and 5‰), all held at 5 °C. This was to identify the transcriptional effects of low salinity at the ambient temperature of the fjord, from which the animals were sourced. At 5 °C, differential gene expression increased with decreasing salinity, with the lowest salinity (5‰) revealing a massive upregulation of gene transcripts compared to the control (17,267 gene transcripts). This was reflected in the GO enrichment analysis, which revealed an increase in the number of upregulated enriched categories as salinity decreased (Figure 1; Appendix A). Cell Adhesion (Biological Process) was one of the most significantly enriched categories at both low and very low salinity when compared with control samples (Figure 1). Nine genes were shared across both low salinity treatment comparisons (Appendix A). These included calcium-dependant cadherins (e.g., *PCDH9* and *FAT4*), which can function as transmembrane proteins connecting the actin cytoskeleton indirectly to neighbouring cells [71], and ependymin-related proteins (e.g., *EPDR1*), a diverse multifunctional gene family, some members of which have been shown to be upregulated in response to environmental stress in bivalves and gastropods [72]. At very low salinity (5‰), five GO enrichment categories were significantly upregulated compared to the control. This included the GO term category Cytoskeleton (Cellular Component), reflecting the most significantly upregulated gene, *Profilin* (log_2_fold change 11.57; Adjusted *p* value 6.87 × 10^−46^), which encodes an actin-binding protein involved in cell shape changes and cytoskeleton rearrangement [73] (Appendix A). Directly comparing very low salinity (5‰) vs. low salinity (15‰) revealed five significantly enriched GO categories, with DNA-binding Transcription Factor Activity (Molecular Process) being the most significant, which was also the case in the 5‰ vs. control comparison. Twenty-four genes were shared across both treatment comparisons for DNA-binding Transcription Factor Activity (e.g., *CREBL2* and *Kat6b*) (Appendix A). Genes identified as being specific to osmoregulatory processes revealed high levels of expression at very low salinity (5‰ vs. control) and when very low salinity and low salinity (5‰ vs. 15‰) were directly compared, but there were a much lower number of upregulated gene transcripts for osmoregulatory regulated genes at low salinity (15‰ vs. control) (Table 3; Figure 2; Appendix A).

### 3.4. Heat Stress and Low Salinity Exposure

These analyses concentrated on mussels exposed to two salinities (23‰ and 15‰), with each subjected to heat shocks of 30 °C and 33 °C during air exposure (control conditions were 23‰ and 5 °C). Differential gene expression patterns under both heat stress treatments (30 °C and 33 °C at 23‰) compared to the control (5 °C at 23‰) revealed similar numbers of upregulated genes (~7000 gene transcripts) at each temperature, while for downregulated genes there was over double the number at 33 °C vs. control than at 30 °C vs. control (Table 2). However, under low salinity exposure (15‰), 4722 gene transcripts were upregulated at 30 °C vs. low salinity (30 °C at 15‰ vs. 5 °C at 15‰), whereas there were 10,796 upregulated at 33 °C vs. low salinity (33 °C at 15‰ vs. 5 °C at 15‰), representing a 129% difference and indicating the synergistic effects of salinity and heat stress (Table 2).

GO term enrichment analysis revealed significant upregulated gene enrichment under both heat stress treatments compared to the control (Figure 3; Appendix A). Cell Morphogenesis (Biological Process) was the most significantly enriched GO term, with 40% of genes shared under both conditions (30 °C vs. control and 33 °C vs. control; Appendix A). A closer inspection of the putative genes involved revealed they encoded heat shock proteins/chaperones (e.g., *HSP70*) (Appendix A). Under the 30 °C treatment, Cell Morphogenesis (Biological Process) was the only significantly enriched category; however, at the higher temperature of 33 °C, eight additional categories were enriched (Figure 3; Appendix A). Those defined as Molecular Function GO categories included DNA-binding Transcription Factor Activity, Enzyme Binding and Unfolded Protein Binding, with genes identified as being involved in transcription activity (e.g., *AP-1*, *CREBL2*, *ATF-2*), programmed cell death (e.g., *Ces-2*) and molecular chaperone activities (e.g., *HSP70*, *HSP90*, *CCT3*) (Appendix A). There was no gene enrichment when heat stress treatments were directly compared at control salinity (33 °C vs. 30 °C both at 23‰), or between the same temperatures at different salinities (Figure 3), indicating that similar processes were upregulated in each treatment. However, when heat stress treatments were compared at lower salinity (low salinity + 33 °C vs. low salinity + 30 °C), seven GO terms were enriched (Figure 3; Appendix A), further indicating the synergistic effects of combined heat and salinity stress. When exposed to low salinity (15‰), there was a larger number of significantly enriched GO terms for each heat stress treatment compared to control salinity (five categories at low salinity + 30 °C compared to one at 30 °C; 19 categories at low salinity + 33 °C compared to nine at 33 °C), with Cell Morphogenesis (Biological Process) again being the most significant (Figure 3; Appendix A). When comparing both 33 °C treatments (15‰ and 23‰) with their respective 5 °C treatments, nine GO enrichment categories were shared (Figure 3; Appendix A), with Cell Morphogenesis (Biological Process) and DNA-binding Transcription Factor Activity (Molecular Function) both being highly significant. Shared genes in the enriched categories relate to stress responses, ubiquitination and protein folding (e.g., *HSP70*, *HSPA5* and *UBC*), transcription activity genes (e.g., *FOSL2*, *CREBL2*) and genes involved in protein syntheses such as elongation factors (e.g., *EEF2*) (Appendix A). Interestingly, although GO enrichment analysis did not reveal specific terms related to environmental stress, many of the genes underlying the significantly upregulated enriched categories (i.e., Cell Morphogenesis and Unfolded Protein Binding) are classically associated with stress responses (e.g., *HSP70*) (Appendix A).

### 3.5. Stress-Related Genes

Datamining of the differentially expressed gene lists (under the Swiss-Prot and NR database annotations) using terms related to oxidative stress, apoptosis, and osmotic and heat shock stress responses (keywords: caspase, catalase, superoxide dismutase, thioredoxin, glutathione, hypoxia, p38 and heat shock) highlighted the clear induction of candidate heat shock genes under the different treatments (Table 4; Appendix A). Interestingly, the highest number of heat shock gene transcripts were found in the very low salinity treatment compared to the control (202 transcripts in total). Furthermore, the majority of identified heat shock gene transcripts in each treatment comparison showed highest sequence similarity to *HSP70* family genes, specifically *HSPA12A* and *HSPA12B*. Comparing the differentially expressed gene transcripts (utilising the SwissProt description) for independent effects of heat stress (30 °C and 33 °C vs. control) and independent effects of low salinity exposure (15‰ and 5‰ vs. control) revealed 143 unique *HSPA12A* unigene transcripts and 119 unique *HSPA12B* unigene transcripts (Appendix A). Less pronounced was the number of transcripts related to classic stress response enzymes associated with oxidative stress and hypoxia (e.g., catalase, superoxide dismutase, thioredoxin and glutathione gene families). Interestingly, there were no upregulated transcripts annotated as p38 mitogen-activated protein kinases, which are considered to play an important role in hypoosmotic stress signalling [74,75]. Gene transcripts related to programmed cell death (e.g., caspase related genes) showed the highest expression in the very low salinity treatment compared to the control (51 transcripts). To validate the transcriptome data, a sample of primers utilised in Nielsen et al. (2021) [16] to amplify stress genes (*HSP70* and *HSP90*) was mapped against the transcriptome generated in this study. These formed strong alignments with a number of gene transcripts (>95% identical sites) (Appendix A). Matching transcripts were checked against the differential expression gene lists in this study and demonstrated similar gene expression patterns, as observed by Nielsen et al. (2021) [16] (Appendix A).

### 3.6. Aquaporin Genes and Phylogenetic Analysis

Within the low salinity treatment comparisons (5‰ vs. control; 15‰ vs. control; 5‰ vs. 15‰), three putative aquaporin gene transcripts were identified in the differentially expressed gene lists (Table 3; Figure 2; Appendix A). Aquaporin channels comprise numerous functionally distinct family members. These have been traditionally divided into four major functional groups, AQP1-like, AQP3-like, AQP8-like and AQP11-like types, based on their distinct evolutionary origin [76]. Previous RNA-Seq studies have successfully used phylogenetic analysis to evaluate the origin of putative aquaporin transcripts identified in a transcriptome (e.g., Calcino et al., 2019 [77]), which provides greater confidence of gene family member identity, compared to blast sequence similarity searches. Phylogenetic analysis utilising the downloaded GenBank aquaporin sequences in combination with the three identified in this study has resolved the four main aquaporin types (AQP1-like, AQP3-like, AQP8-like and AQP11-like) into four well supported clades (Bootstrap values (BS) > 90%). Two of the three putative aquaporin transcripts identified formed a clade with the AQP8-like aquaporins (97% BS), while the third clustered within the AQP1-like aquaporin clade (Figure 4).

## 4. Discussion

Previous work investigating the effects of heat and osmotic stress on *M. edulis* survival demonstrated that exposure to reduced salinity decreased the temperature at which mortality occurred (see Figure 2 in Nielsen et al. (2021) [16]). The molecular results presented here offer insight into the underlying mechanisms that account for these observed outcomes. The results for the effects of increased air temperature on *M. edulis* confirm the high degree of thermal tolerance previously observed by Clark et al. (2021) [17], who examined the effect of temperatures up to 32 °C on *M. edulis* using RNA-Seq. In the current study, significant enrichment for the cellular stress response (CSR), in particular apoptosis-related functions, was not observed. However, the presence of multiple heat shock genes, revealed through differentially expressed datamining (Table 4) and within enriched GO terms at 33 °C (e.g., Cell Morphogenesis and Unfolded Protein Binding), suggested that stress pathways were clearly active. Furthermore, there was no GO term enrichment when directly comparing the 33 °C and 30 °C treatments. This reflected the lack of mortality observed in Nielsen et al. (2021) [16] in mussels exposed to heat shock at both 30 °C and 33 °C.

The molecular response to low salinity (15‰) was indicative of at least partial acclimation to low salinity, evidenced by the relative low number of upregulated genes (particularly stress and osmoregulatory genes) and low number of enriched GO terms, reflecting the 100% survival rates after 6 days of exposure [16]. *M. edulis* has an extremely wide estuarine and marine distribution [19], and the molecular mechanisms revealed here demonstrate a remarkable degree of efficiency at dealing with osmotic stress, especially considering that the mussels used in this experiment were already acclimated to a salinity (23‰) significantly below full-strength sea water that averages 35‰. This degree of euryhalinity is likely reflective of acclimatisation to the summer environmental conditions experienced by *M. edulis* in the inner Greenland fjords where they were collected [16,78].

In mussels exposed to low salinity, the effects of heat shock increased the overall gene expression (3781 gene transcripts upregulated at 30 °C at 15‰ and 6243 upregulated at 33 °C at 15‰; Table 2), including increased numbers of stress genes (67 upregulated at 30 °C at 15‰ and 86 upregulated at 33 °C at 15‰; Table 4), compared to the effects of heat shock alone (30 °C at 23‰ and 33 °C at 23‰). Furthermore, directly comparing the heat-stressed treatments at low salinity (low salinity + 33 °C vs. low salinity + 30 °C) revealed seven significantly enriched categories, while there was no enrichment when directly comparing 33 °C and 30 °C at control salinity. The mortality data for these treatments (low salinity + 30 °C/33 °C) from Nielsen et al. (2021) [16] show a small increase in mortality at 30 °C increasing to around 20% of the sample population at 33 °C. Taken together, this indicates that accommodating both environmental stressors represents a tipping point in the mussels’ ability to cope with stress. At very low salinity (5‰), that threshold has clearly been breached, evidenced by the dramatic rise in mortality as heat stress was applied [16].

When multiple environmental stressors are applied at the same time, they may interact in complex, non-additive ways (e.g., synergistic or antagonistic) resulting in outcomes that are difficult to predict from one stressor alone [79]. Additionally, each environmental stressor may impact differing molecular and physiological pathways of an organism, resulting in an energy allocation trade-off between competing demands for metabolic functions [40]. For example, cell volume regulation in response to reduced salinity is metabolically costly [31]. At the same time, organisms remodel the composition of membrane phospholipids in response to salinity and temperature changes using different mechanisms [80,81], while the energy required to maintain basal metabolic rate, on average, increases with temperature [1,2,4]. Therefore, when both stressors are applied in combination, metabolic resources may become limiting, and a trade-off occurs. Previous studies comparing differential gene expression in *Mytilus* species under both high temperature and low salinity observed only a small number of overlapping genes, which responded in divergent expression patterns [43,44]. In contrast, in the current study, directly comparing differentially expressed gene transcripts between low salinity compared to control (15‰ at 5 °C vs. 23‰ at 5 °C) and 30 °C heat stress compared to control (30 °C at 23‰ vs. 5 °C at 23‰) revealed 1148 upregulated and 1072 downregulated shared gene transcripts (Appendix A), suggesting that there was a substantial number of molecular responses shared by each independent stressor. It is important to note that Lockwood and Somero (2011) [44] measured the acute response to hypo-osmotic shock (over a four-hour immersion), while the reduction in salinity was from 35‰ to 29.75‰. In addition, the use of microarrays in their experiment constrained the number of genes examined, potentially obscuring differentially expressed genes, particularly from closely related gene family members. These are all factors likely accounting for the discrepancy in outcomes between Lockwood and Somero (2011) [44] and the current study. Our study clearly showed that the combination of stressors appeared to work in a synergistic fashion on a gene expression level, with gene upregulation more than doubling at the highest stressor combination (low salinity + 33 °C vs. low salinity) compared to the effects of the slightly lower heat stress (low salinity + 30 °C vs. low salinity). As competition for resource allocation between osmoregulation and heat stress was stretched, additional stress responses appeared to be triggered, perhaps in a positive feedback-type scenario. Ultimately, the drain on energy resources appeared to reduce the capacity for *M. edulis* to tolerate heat stress. However, the large overlap in genes responsible for salinity tolerance and heat shock suggested that both stressors stimulate similar components of the CSR. The level of macromolecular damage rather than the specificity of the abiotic stressor is considered to be the trigger for the CSR [82,83], meaning that pre-conditioning to one stressor may confer an increased tolerance to another, a phenomenon known as cross-tolerance [84,85,86]. Pre-conditioning was not a component of the current study; however, testing the hypothesis that an osmotic shock and subsequent recovery may offer increased tolerance to heat shock in *M. edulis* would be an interesting avenue for future researchers.

### 4.1. HSPA12 Genes

An interesting finding was the lack of GO term enrichment for HSPs or the CSR when heat stress treatments were compared at different salinities (Figure 3). However, there was high expression of a set of HSP70 family genes, *HSPA12*, upregulated under every treatment. *HSPA12* genes are structurally and functionally distinct from the classical HSP70 family members, with phylogenetic analysis implying that they have highly divergent origins from other *HSP70*s [87,88,89]. Clark et al. (2021) [17] found a similar expansion of *HSPA12* genes and a concurrent lack of enrichment for a cellular stress response in *M. edulis* upon heat shock. Clark et al. (2021) [17] suggested that through the process of gene duplication events, multiple copies of HSPA12 proteins have taken on subtly differing roles and collectively act as intertidal stress regulators, offering substantial thermal tolerance. Enhanced expression of *HSPA12* genes in other marine bivalves suggests that particular sub-sets of *HSPA12* are stressor-specific, with variations in expression levels demonstrated in response to toxic shock and heavy-metal contamination [90,91]. Indeed, in the current study large numbers of unique *HSPA12* gene transcripts were identified under both heat stress and hypoosmotic exposure. For example, out of 143 identified *HSPA12A* gene transcripts, 23 were uniquely differentially expressed as a response to heat stress, 45 for hypoosmotic stress and 25 were shared between stressors (Appendix A). These results highlight the specific role that *HSPA12* may undertake in response to osmotic stress and imply that their stressor specific sub-functionalisation is a key molecular adaptation enabling stress tolerance in extreme environments.

### 4.2. Molecular Response to Low Salinity Exposure

The finding that low salinity reduces the thermal resilience of *M. edulis* focussed the current analyses on the molecular responses to osmotic stress. The molecular data from *M. edulis* exposed to low salinity at 5 °C demonstrated changes for processes involved in osmoregulation and cell volume regulation. At the lowest salinity, there was significant upregulation of genes putatively involved in active membrane transport such as ATPase pumps (e.g., *ATP2B3* and *nkb-3*), potassium and chloride ion membrane channels (e.g., *KOR2*, *Sh*, *Kcnk10*, *CLIC6*, *CLCD*) and ion exchangers (e.g., *SLC9B2* and *SLC8A1*), indicating a change in osmoregulatory response (Table 3; Figure 2; Appendix A). In addition, there was also significant upregulation of organic osmolyte transporter genes (e.g., *SLC6A5* and *SLC6A9*), suggesting an increased demand to balance solute concentrations. As an osmoconformer, *M. edulis* is considered to have little ability to regulate extracellular fluid osmolality when exposed to decreasing salinity [23,92]. Instead, to balance homeostasis within the cell, mussels reduce cytosol solute concentrations to match that of the external environment, primarily through adjustments in the intracellular concentration of free amino acids and other small organic osmolytes [30,93]. The inorganic ion proportion of intracellular osmolality (e.g., K^+^, Na^+^, Cl^−^) is suggested to remain at a relatively constant level (~500–600 mOsm kg^−1^ within marine animals [94]). In the 5‰ treatment, the cytosol of *M. edulis* was likely to be highly hyperosmotic in comparison to the external haemolymph, which would be expected to be near isosmotic with the external low salinity seawater due to the lack of extracellular osmoregulatory abilities [23,92]. To maintain the minimum level of ions and organic osmolytes necessary for cellular functions, active uptake from the extracellular fluids is necessary. This was reflected by the increased upregulation at 5‰ (5‰ vs. control) of ATPase genes including *ATP2B3* (log_2_fold change 6.12; Adjusted *p* value 2.75 × 10^−4^)*,* and osmolyte transporter genes including sodium- and chloride-dependent glycine transporter gene, *SLC6A5* (log_2_fold change 6.15; Adjusted *p* value 7.33 × 10^−6^), which had been previously identified as a marker gene for the effects of low salinity [45]. Lockwood and Somero (2011) [44] investigated low salinity stress in two *Mytilus* species, *M. galloprovincialis* and *M. trossulus*, and observed upregulation of ion channels but downregulation of ion and amino acid transporters in response to an acute low salinity (29.75‰) exposure over four hours. This response was considered to reflect the need to allow ion concentrations to equalise, while reducing active transport of ions and osmolytes into the cell, which would be hyperosmotic to the external medium. In contrast, the duration of osmotic stress in our study was six days, akin to the beginning of an acclimation process, which has been proposed to take between one and four weeks in temperate marine animals [32]. In addition, the *Mytilus* species, *M. galloprovincialis* and *M. trossulus* in Lockwood and Somero (2011) [44], were originally acclimated to 35‰ and then exposed to only a four-hour immersion at ~30‰ prior to sampling, representing a 15% change in osmotic pressure. *M. edulis* in the current study was already acclimated to 23‰ in the natural environment and then experienced a ~35% and ~78% change in osmotic pressure when exposed for six days at 15‰ and 5‰ salinity respectively. Therefore, it is not surprising to find considerable differences in transcriptomic response between these two sets of low salinity experiments, with the former assessing the acute response, while this study is a longer-term acclimation experiment, with larger changes in external salinity. Previous investigations into the long-term acclimation to salinity stress in the blue crab, *Callinectes sapidus,* demonstrated an increase in the activity and synthesis of ATPase pumps in posterior gill tissue [95]. In the current experiment, upregulation of a number of ATPase pumps at 5‰ salinity is consistent with these findings.

After six days of exposure to 15‰ salinity, there was relatively little differential expression of genes related to ion channels (15‰ vs. control) in comparison to 5‰ salinity (5‰ vs. control) (Table 3). At 15‰ salinity (15‰ vs. control), one copy of a potassium voltage-gated channel gene (*Sh*) and one potassium channel gene (*KOR2*) were downregulated; however, one calcium-activated potassium channel gene (*orai1*) and one potassium channel gene (*SK*) were upregulated (Figure 2; Appendix A). Interestingly, the same potassium voltage-gated channel gene, *Sh*, that was downregulated at 15‰ salinity (15‰ vs. control) was significantly upregulated in the 5‰ vs. 15‰ salinity treatment. The activation of voltage-gated channels is considered to occur within milliseconds upon membrane depolarisation [96], allowing for rapid osmotic re-balancing. The downregulation of a potassium voltage-gated channel after six days exposure at 15‰ suggests that membrane depolarisation has led to a stable change in osmotic balance. Meng et al. (2013) [45] observed a similar expression pattern in the Pacific oyster, *Magallana gigas*, after seven days exposure to 15‰ salinity. As 15‰ salinity is a value comparable to the ionic osmolality of the cell cytosol (~ 500–600 mOsm kg^−1^ [94]), presumably adjustments to balance the osmotic gradient may have already taken place over a shorter time frame, and at the point of sampling there was no requirement for an osmoregulatory adjustment. This is further validated by the relatively small number of classical stress response genes upregulated at 15‰ (15‰ vs. control) in this study, indicating that *M. edulis* has partially acclimated or is near to acclimating to the lower salinity.

At 5‰ salinity (5‰ vs. control), the increased upregulation of potassium and chloride channel proteins in addition to ion exchangers and osmolyte transporters likely reflected the ongoing need to maintain the intracellular environment as hyperosmotic to the haemolymph. At the same time, water would be osmotically drawn into the cytosol. Aquaporins are membrane channel proteins that allow for the passive transport of water and uncharged solutes [97,98] and are considered to play an essential role in osmoregulation in molluscs [76]. Interestingly, there was significant upregulation of an aquaporin transcript at 5‰ salinity (5‰ vs. control) and two when 5‰ and 15‰ treatments were compared. Increased aquaporin production within the external cellular membrane would enable passive water flow into the cell, which would appear counterproductive, considering the cell is likely already dealing with excess intracellular water. In the freshwater mussel, *Dreissena rostriformis*, recent evidence suggests that during embryogenesis, aquaporins, vacuolar-type ATPase (V-ATPase) and sodium/hydrogen exchangers are involved in osmoregulation in low salinity environments, and were likely a key evolutionary step in the transition from marine to freshwater tolerance [77]. Calcino et al. (2019) [77] described the formation of embryonic ‘cleavage cavities’, which fill with excess cellular water that is later discharged outside the cell. These membrane-bound cavities are acidified by V-ATPase pumps, creating an internal solute concentration higher than the rest of the cell cytosol. Aquaporin water channels facilitate the movement of water down osmotic gradients into the swelling cavity, which is later expelled at the cell membrane, possibly in a manner similar to contractile vacuoles in protists [99,100]. Phylogenetic analysis of the putative aquaporin transcripts identified in the low salinity exposure analysis from the current study showed clustering within both AQP1-like and AQP8-like clades for the upregulated aquaporins. AQP1-like aquaporins are considered to be classical aquaporins, allowing for selective uptake of water molecules, while AQP8-like aquaporins are ammonia-selective and have been termed aquaamoniaporins [76]. Interestingly, in vertebrate hepatocytes it has been demonstrated that AQP8 plays a significant role in water transport during bile secretion, where it is localised in intracellular vesicles [100]. Upon cAMP signalling, AQP8-associated vesicles can be rapidly inserted into the cell membrane between cells, facilitating the movement of water to form a fluid-filled structure called the canaliculus, which has been suggested to be very similar in form to the cleavage cavities observed in *D. rostriformis* [77]. At 15‰ salinity (15‰ vs. control), there was downregulation of a separate aquaporin identified by the phylogeny as ATP8-like, which interestingly showed neutral expression at 5‰ (5‰ vs. control). This implies that these particular aquaporins were located in the external cell membrane. By reducing their synthesis while exposed to low salinity (15‰), the cell minimises passive water movement into the cytosol to prevent excess swelling, and cell volume regulation is undertaken. Very similar RNA-Seq results were observed in the Pacific oyster, *M. gigas*, after seven days exposure to low salinity, where the lowest expression for three aquaporin mRNA was at 15‰ salinity [45].

In this study, there was upregulation of putative V-ATPase transcripts (e.g., *Atp6v0a1*, *ATP6AP1*) at 5‰ salinity (5‰ vs. control), and three more (*VhaAC39-1*, *Vha26*, *Atp6v0a1*) when 5‰ and 15‰ treatments were compared. In addition, sodium/hydrogen and sodium/calcium exchangers were upregulated at 5‰ salinity (5‰ vs. control) (e.g., *SLC9B2* and *SLC8A1*), suggesting the need to recover sodium from the extracellular fluid. Together, the finding of increased expression of aquaporins, V-ATPase and ion exchangers at very low salinity suggested that *M. edulis* may use a similar method for coping with the constant influx of osmotically driven water at very low salinities as observed in embryotic freshwater bivalves and fluid transport in more complex adult vertebrate organs [77,100].

### 4.3. Cytoskeleton

The actin cytoskeleton is considered to play an essential role in cell volume regulation under osmotic stress [101]. For cells in a hypotonic environment, as in the case of very low salinity, the influx of osmotically drawn water causes the cell to swell, which can result in extensive re-organisation of the actin cytoskeleton [101]. Within the current study, many significantly upregulated genes (including *Profilin*) and a number of GO terms (Cytoskeleton and Cell Adhesion) relating to the actin cytoskeleton were identified. This suggests that the long-term effects of low salinity exposure induced structural changes to the cell cytoskeleton to counter the effects of maintaining homeostasis in a hypotonic environment. Previous studies have identified upregulation of *Profilin* in response to osmotic stress in annelids [102], in addition to both oxidative and heavy-metal stress in *M. galloprovincialis* [103,104]. Osmotic stress is also associated with specific heat shock proteins, in particular small heat shock proteins (sHSPs), which are known to colocalise with the cell cytoskeleton when under osmotic stress [105]. Our data show significant upregulation in sHSPs under very low salinity (5‰ vs. control), while there was almost none at 15‰, compared to control salinity. Previous studies on renal and ocular physiology have shown that sHSPs are synthesised on exposure to high NaCl concentrations and remain at high levels as long as cells are exposed to osmotic stress [106,107]. Beck et al. (2000) [108] suggested that it is the tonicity not the osmolality that is responsible for the presence of sHSPs, which likely play a role in acclimation to differing salinities by their involvement in cytoskeleton modulation. It is therefore possible that the high expression of sHSPs at very low salinity (5‰) in our data suggests the cell is under constant cellular stress as a result of swelling due to the hypotonic environment, while their almost complete absence at 15‰ implies that the cell osmolality is in balance with the external medium and therefore acclimation is near completion.

## 5. Conclusions

The molecular data within this study confirm that *M. edulis* has considerable resilience to heat stress and highly efficient mechanisms to acclimatise to lowered salinity. Thermal and osmotic stress tolerance appear to be regulated by stressor-specific sets of *HSPA12* genes, which may function as intertidal stress regulators, as previously suggested [17]. However, although both stressors also induce several shared molecular responses, when they are applied in combination their synergistic effects can push the animal to the edge of their tolerance thresholds before either factor alone. This implies that, although acclimation to lower salinity (15‰) is possible, the underlying energetic cost for maintaining homeostasis appears to be greater than at pre-acclimation levels, which may help account for the observed impacts on growth and morphology in *M. edulis* living in lower salinity environments [34,36,38]. At very low salinities, the molecular data suggested *M. edulis* is in a constant state of stress and appeared to be activating an osmoregulatory response to maintain a hyperosmotic intracellular environment while also implementing structural changes to the cell cytoskeleton. We suggest that *M. edulis* may use a similar mechanism to excrete excess intracellular water to that found in embryonic freshwater mussels and more complex adult vertebrate organs [77,100].

The current study highlights the importance of combined stressor studies for predicting the future range abundance and distribution of Arctic intertidal species. Furthermore, it is also of clear value for the mussel aquaculture industry when identifying future sites for commercial ventures in view of predicted environmental changes. Overall, these results inform our understanding of how *M. edulis* responds at the molecular level to the effects of heat stress, freshening, and their combined interaction, offering insights into the ability of intertidal mussels to resist future climate changes, and a measure of their physiological limitations.

## Figures and Tables

**Figure 1 genes-13-00155-f001:**
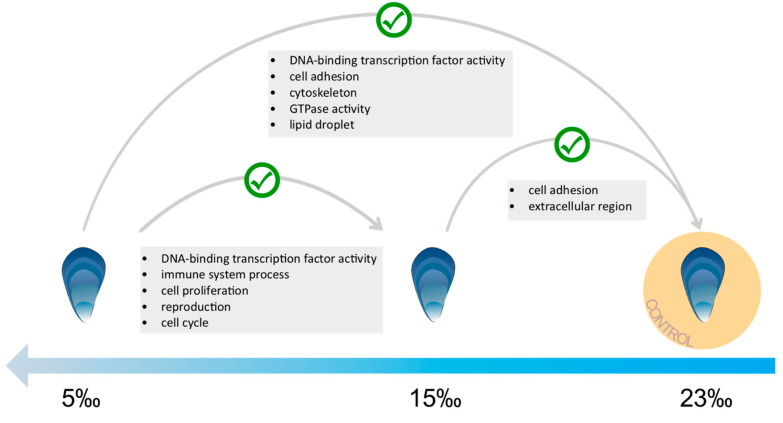
Diagrammatic Gene Ontology term (GO) enrichment results for upregulated gene expression in blue mussels (*M. edulis*) from low salinity exposure treatment comparisons. Green tick represents enrichment between treatments. Lists within grey boxes are significantly enriched GO term categories representing functional profiles of the overexpressed set of genes from the sample comparisons (See Appendix A).

**Figure 2 genes-13-00155-f002:**
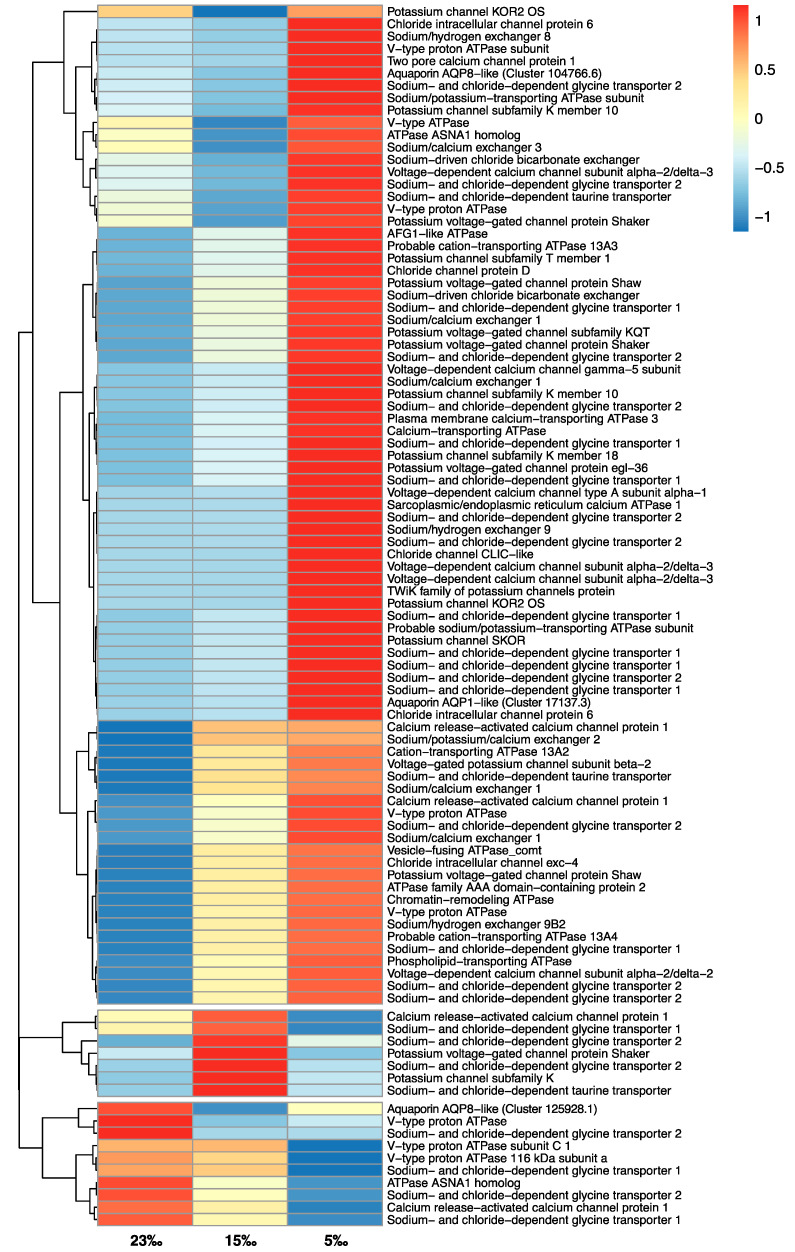
Heatmap showing relative fold change expression of differentially expressed putative osmoregulatory genes in blue mussels (*M. edulis*) from low salinity exposure treatment comparisons, with SwissProt gene transcript descriptions (see Appendix A for transcript identification and FPKM values). Colour scale bar indicates relative expression: red = upregulation; blue = downregulation.

**Figure 3 genes-13-00155-f003:**
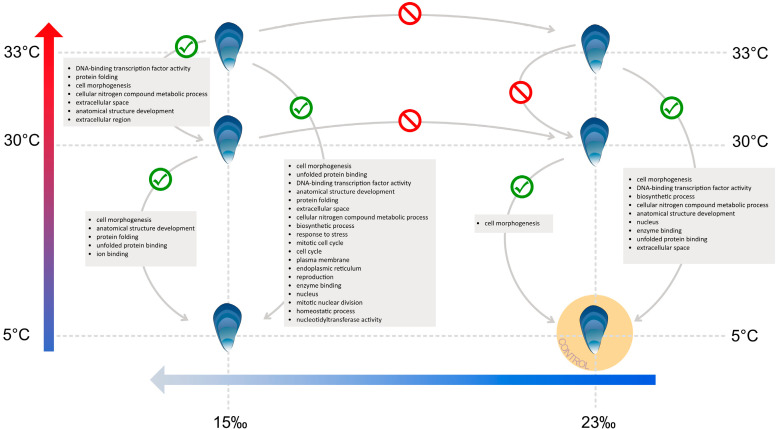
Diagrammatic gene ontology (GO) term enrichment results for upregulated gene expression in blue mussels (*M. edulis*) from heat stress and low salinity exposure treatments. Green tick represents enrichment between treatments. Red no symbol indicates a lack of significant gene enrichment. Lists within grey boxes are significantly enriched GO term categories representing functional profiles of the overexpressed set of genes from the sample comparisons (See Appendix A).

**Figure 4 genes-13-00155-f004:**
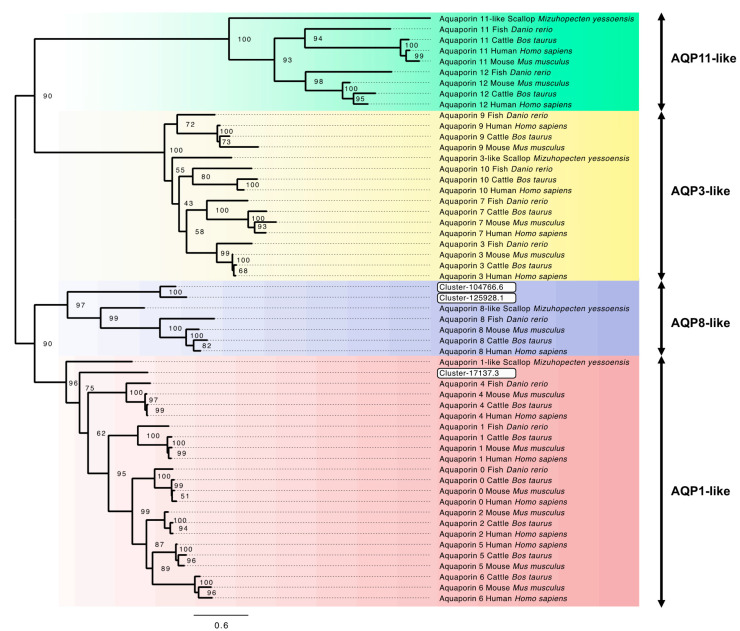
Maximum likelihood unrooted phylogeny of aquaporin genes *AQP0-12* from GenBank including human, mouse, cattle, fish and mollusc (for GenBank assession numbers, see Appendix A), and blue mussel (*M. edulis*) putative aquaporins transcripts identified in this study (highlighted in white boxes). Bootstrap values at nodes. *AQP2*, *AQP5* and *AQP6* orthologues are absent in the zebrafish *Danio rerio*. The scallop, *Mizuhopecten yessoensis* aquaporin genes are predicted from genome analysis; therefore, only a single sequence representative from each for the four aquaporin classes has been included.

**Table 1 genes-13-00155-t001:** Blue mussel (*M. edulis*) transcriptome assembly statistics and gene annotation rates.

Assembly Statistics	
Number of nucleotides (nt)	394,372,995
Number of unigenes	449,638
Unigene minimum length (bp)	301
Unigene mean length (bp)	877
Unigene medium length (bp)	621
Unigene maximum length (bp)	17,583
N50	1104
N90	420
Number of unigenes between 200–500 bp	165,273
Number of unigenes between 500–1k bp	163,324
Number of unigenes between 1k–2k bp	89,465
**Annotation Rates**	
Database	% annotation
NR (NCBI non-redundant protein sequences)	35.08
NT (NCBI nucleotide sequences)	9.75
KEGG (Kyoto Encyclopaedia of Genes and Genomes Orthologues)	5.4
Swiss-Prot: (Manually annotated protein sequences)	19.35
Pfam: (Protein domains and families)	12.33
GO: (Gene Ontology)	9.54
KOG: (euKaryotic Orthologous Groups)	7.96
Annotated in all Databases	0.51
Annotated in at least one Database	40.68

**Table 2 genes-13-00155-t002:** Differential gene expression counts in blue mussels (*M. edulis*) subjected to different heat and osmotic stress treatments. Treatment descriptions are as follows: control (23‰ at 5 °C), low salinity (15‰ at 5 °C), very low salinity (5‰ at 5 °C), 30 °C (23‰ at 30 °C), 33 °C (23‰ at 33 °C), low salinity + 30 °C (15‰ at 30 °C), and low salinity + 33 °C (15‰ at 33 °C).

Comparison	Upregulated Genes	Downregulated Genes	Total Genes
**Low salinity exposure**			
Low salinity vs. control	5004	3768	8772
Very low salinity vs. control	17,267	6824	24,091
Very low salinity vs. low salinity	10,466	5354	15,820
**Heat stress + low salinity exposure**			
30 °C vs. control	7002	4195	11,197
33 °C vs. control	7033	8824	15,857
33 °C vs. 30 °C	3958	8314	12,272
Low salinity + 30 °C vs. 30 °C	3781	4734	8515
Low salinity + 33 °C vs. 33 °C	6243	2802	9045
Low salinity + 33 °C vs. low salinity + 30 °C	7506	4948	12,454
Low salinity + 30 °C vs. low salinity	4722	4540	9262
Low salinity + 33 °C vs. low salinity	10,796	7202	17,998

**Table 3 genes-13-00155-t003:** Osmoregulatory marker genes identified in the differentially expressed profiles from low salinity exposure treatment comparisons, using the SwissProt annotations (see Appendix A for gene transcript lists). ↑: represents upregulated genes, ↓: represents downregulated genes.

Gene Type	Comparisons					
	Low Salinity vs. Control		Very Low Salinity vs. Control		Very Low Salinity vs. Low Salinity	
	↑	↓	↑	↓	↑	↓
Ion channel: Potassium channel	2	2	14	0	5	1
Ion channel: Chloride channel	0	0	4	0	2	0
Ion channel: Calcium channel	1	0	6	2	4	1
Ion exchangers	1	1	8	0	4	0
Aquaporins	0	1	1	0	2	0
ATPase pumps (V-type)	0	2	3	2	3	0
ATPase pumps (Other)	0	0	11	1	3	0
Osmolyte transporters (*SLC6A5*/*SLC6A9*)	4	0	17	5	7	0

**Table 4 genes-13-00155-t004:** Upregulated stress response genes in blue mussels (*M. edulis*) identified in differentially expressed gene lists from the different treatment comparison, using the SwissProt and NR database annotations.

Gene Family	Family Member	Comparisons								
		30 °C vs. Control	33 °C vs. Control	Low Sal. + 30 °C vs. 30 °C	Low Sal. + 33 °C vs. 33 °C	Low Sal. + 30 °C vs. Low Salinity	Low Sal. + 33 °Cvs. Low Salinity	Low Salinity vs. Control	Very Low Salinity vs. Control	Very Low Salinity vs. Low Salinity
Catalase		0	0	0	0	0	0	0	0	0
Caspase		16	8	9	13	10	16	14	51	39
Glutathione		12	8	7	13	5	15	4	14	11
Hypoxia		1	0	0	2	0	1	0	2	1
Superoxide dismutase		4	8	5	6	4	11	4	5	1
Thioredoxin		5	4	3	3	2	7	7	11	1
p38 mitogen-activated protein kinases		0	0	0	0	0	0	0	0	0
Heat shock proteins	HSPA12A	27	11	15	21	17	21	18	61	54
	HSPA12B	12	9	20	20	19	22	9	59	48
	small HSP	24	24	0	1	22	23	1	22	13
	HSP68	23	23	1	1	24	22	0	19	16
	HSPA5	1	10	0	0	7	12	0	0	0
	HSP70B2	8	11	2	1	9	10	0	9	9
	HSC70	2	4	1	0	6	5	1	3	7
	HSP90	0	0	0	1	7	10	1	1	3
	HSP110	1	0	0	0	1	1	0	0	0
	HSP70 Family	39	29	4	4	31	36	1	28	27
	HSP total	137	121	43	49	143	162	31	202	177
All stress genes		175	149	67	86	164	212	60	285	230

## Data Availability

RNA-Seq data has been deposited in the ArrayExpress database at EMBL-EBI (https://www.ebi.ac.uk/arrayexpress/experiments/E-MTAB-11178) (accessed on 17 December 2021) under accession number E-MTAB-11178. The assembled transcriptome has been deposited in the European Nucleotide Archive (ENA) under accession number ERZ4783392 accessible via https://www.ebi.ac.uk/ena/browser/view/PRJEB48959 (accessed on 17 December 2021).

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
