# Peer review of "Molecular Responses to Thermal and Osmotic Stress in Arctic Intertidal Mussels (Mytilus edulis): The Limits of Resilience"

_genes, 2022, doi:10.3390/genes13010155_

Round 1
Reviewer 1 Report
In this study, the authors well designed the experiment and performed it to investigate the molecular response of mussels under salinity and temperature stresses. Also, the manuscript was well-written and its results were able to answer the questions mentioned in introduction part. There are some further comments for authors and the authors should improve the manuscript before final decision by the editors.
- In the introduction, the authors introduced the response of Mytilus edulis to the salinity stress based on the related research. However, the response of this species to the temperature stress was mentioned shortly. Authors should add more information about the response of Mytilus edulis under the temperature variation.
- The authors analyzed the aquaporin gene as an osmoregulatory marker and reconstructed phylogeny for different organisms based on sequence comparison of the gene. However, the manuscript did not describe the reason why phylogenetic analysis of aquaporin is important for this manuscript. the authors should provide more explanation about this.
- In the conclusion, the authors should add suggestions utilizing of this study such as aquaculture for further studies
- For temperature experiment, authors treated Mytilus edulis at 30 °C and 33 °C. The temperature range is small, while Mytilus edulis can withstand high variation of temperature as you described in line 48 to 52. Therefore, the author should provide whether physical process affected by the small range of temperature and changes at the molecular level observed during the temperature treatments.
Author Response
Thank you for your constructive comments, we feel they have enhanced and improved the manuscript. Our responses to your comments are below:
Point 1: In the introduction, the authors introduced the response of Mytilus edulis to the salinity stress based on the related research. However, the response of this species to the temperature stress was mentioned shortly. Authors should add more information about the response of Mytilus edulis under the temperature variation.
Response: We suggest adding the below section after line 52:
Upper lethal temperatures for M. edulis have been observed to be between 25°C and 38°C (Bayne, 1976; Jones et al., 2010; Thyrring et al., 2015, 2019). This high degree of thermal tolerance in Mytilus species is largely dependent on environmental conditions, with acclimation and the predictability of thermal stress being considered key modifiers of thermal resilience (Connor and Gracey, 2020; Clark et al., 2021).
Point 2: The authors analyzed the aquaporin gene as an osmoregulatory marker and reconstructed phylogeny for different organisms based on sequence comparison of the gene. However, the manuscript did not describe the reason why phylogenetic analysis of aquaporin is important for this manuscript. the authors should provide more explanation about this.
Response: We suggest adding the following in the results section 3.6, after the full stop on line 386:
Aquaporin channels comprise numerous functionally distinct family members. These have been traditionally divided into four major functional groups, AQP1-like, AQP3-like, AQP8-like and AQP11-like types, based on their distinct evolutionary origin (Kosicka et al., 2020). Previous RNA-Seq studies have successfully used phylogenetic analysis to evaluate the origin of putative aquaporin transcripts identified in a transcriptome (e.g., Calcino et al., 2019), which provides greater confidence of gene family member identity, compared to blast sequence similarity searches.
Point 3: In the conclusion, the authors should add suggestions utilizing of this study such as aquaculture for further studies
Response: We suggest adding the following before the beginning of the new sentence on line 666:
The current study highlights the importance of combined stressor studies for predicting future range abundance and distribution of Arctic intertidal species. Furthermore, it is also of clear value for the mussel aquaculture industry when identifying future sites for commercial ventures in view of predicted environmental changes.
Point 4: For temperature experiment, authors treated Mytilus edulis at 30 °C and 33 °C. The temperature range is small, while Mytilus edulis can withstand high variation of temperature as you described in line 48 to 52. Therefore, the author should provide whether physical process affected by the small range of temperature and changes at the molecular level observed during the temperature treatments.
Response: As we know the lethal temperature for M. edulis is 36°C under heat stress alone (see Nielsen et al., 2021), we have chosen temperatures close to but below this threshold to investigate the effects of how combined temperature and low salinity affect physiological limits. We suggest adding the following line to the Materials and Methods section 2.2., after the full stop on line 122 (prior sentence is included in blue):
Within each salinity treatment, mussels were exposed to experimental air temperatures of 5°C, 30°C, and 33°C every 12th hour for 1.5 hr over a 6-day period. As M. edulis has an upper temperature limit of 36˚C, temperature values close to the upper limit were chosen.
References:
Bayne, B. L. Marine Mussels, Their Ecology and Physiology; Bayne, B. L., Ed.; Cambridge University Press, 1976.
Calcino, A. D., De Oliveira, A. L., Simakov, O., Schwaha, T., Zieger, E., Wollesen, T., et al. (2019). The quagga mussel genome and the evolution of freshwater tolerance. DNA Res. 26, 411–422. doi:10.1093/dnares/dsz019.
Clark, M. S., Peck, L. S., and Thyrring, J. (2021). Resilience in Greenland intertidal Mytilus: The hidden stress defense. Sci. Total Environ. 767, 144366. doi:10.1016/j.scitotenv.2020.144366.
Connor, K., and Gracey, A. Y. (2020). Cycles of heat and aerial-exposure induce changes in the transcriptome related to cell regulation and metabolism in Mytilus californianus. Mar. Biol. 167, 1–12. doi:10.1007/s00227-020-03750-6.
Jones, S. J., Lima, F. P., and Wethey, D. S. (2010). Rising environmental temperatures and biogeography: Poleward range contraction of the blue mussel, Mytilus edulis L., in the western Atlantic. J. Biogeogr. 37, 2243–2259. doi:10.1111/j.1365-2699.2010.02386.x.
Kosicka, E., Lesicki, A., and Pieńkowska, J. R. (2020). Molluscan aquaporins: an overview, with some notes on their role in the entry into aestivation in gastropods. Molluscan Res. 40, 101–111. doi:10.1080/13235818.2020.1716442.
Thyrring, J., Rysgaard, S., Blicher, M. E., and Sejr, M. K. (2015). Metabolic cold adaptation and aerobic performance of blue mussels (Mytilus edulis) along a temperature gradient into the High Arctic region. Mar. Biol. 162, 235–243. doi:10.1007/s00227-014-2575-7.
Thyrring, J., Tremblay, R., and Sejr, M. K. (2019). Local cold adaption increases the thermal window of temperate mussels in the Arctic. Conserv. Physiol. 7. doi:10.1093/conphys/coz098.
Reviewer 2 Report
The study of the interaction of climatic indices and their effect on M. edulis, as well as the molecular responses it exhibits, highlights an interesting acclimation mechanism that merits further investigation, under the scope of global climate change.
Lines 134-138: Please clarify the assumption that "In the latter, the assumption was made that survivors represented more tolerant individuals and would not be fully representative of the overall population response". How did this assumption affect treatment selection? How is individual tolerance defined?
Additionally, it could be noted that the conclusions session could be further enriched, as the discussion includes interesting results.
Author Response
Thank you for your constructive comments, we feel they have enhanced and improved the manuscript. Our responses to your comments are below:
Point 1: Lines 134-138: Please clarify the assumption that "In the latter, the assumption was made that survivors represented more tolerant individuals and would not be fully representative of the overall population response". How did this assumption affect treatment selection? How is individual tolerance defined?
Response: We suggest removing this line from 136:
In the latter, the assumption was made that survivors represented more tolerant individuals and would not be fully representative of the overall population response.
And replacing with the following (prior sentence is included in blue):
The 36°C air temperature treatment resulted in 100% mortality, while mortality in mussels exposed to 5‰ salinity and subjected to heat stress (both 30°C and 33°C) were >50%, representing a median lethal mortality (LD50).
Point 2: Additionally, it could be noted that the conclusions session could be further enriched, as the discussion includes interesting results.
Response: We suggest adding the following between line 654 and 655:
Thermal and osmotic stress tolerance appears to be regulated by stressor specific sets of HSPA12 genes, which may function as intertidal stress regulators as previously suggested (Clark et al., 2021).
And the following after the full stop on line 666:
The current study highlights the importance of combined stressor studies for predicting future range abundance and distribution of Arctic intertidal species. Furthermore, it is also of clear value for the mussel aquaculture industry when identifying future sites for commercial ventures in view of predicted environmental changes.
References
Clark, M. S., Peck, L. S., and Thyrring, J. (2021). Resilience in Greenland intertidal Mytilus: The hidden stress defense. Sci. Total Environ. 767, 144366. doi:10.1016/j.scitotenv.2020.144366.